# 4'-Fluorouridine mitigates lethal infection with pandemic human and highly pathogenic avian influenza viruses

**Carolin M. Lieber[1], Megha Aggarwal[1], Jeong-Joong Yoon[1], Robert M. Cox[1], Hae-Ji Kang[1], Julien Sourimant[1], Mart Toots[1], Scott K. Johnson[2], Cheryl A. Jones[2], Zachary M. Sticher[3], Alexander A. Kolykhalov[3], Manohar T. Saindane[3], Stephen M. Tompkins[2], Oliver Planz[4], George R. Painter[3], Michael G. Natchus[3], Kaori Sakamoto[5], Richard K. Plemper**  **[1] \***

1 Center for Translational Antiviral Research, Georgia State University Institute for Biomedical Sciences, Atlanta, Georgia, United States of America, 2 Center for Vaccines and Immunology, University of Georgia, Athens, Georgia, United States of America, 3 Emory Institute for Drug Development, Emory University, Atlanta, Georgia, United States of America, 4 Department of Immunology, Interfaculty Institute for Cell Biology, Eberhard Karls University Tübingen, Tübingen, Germany, 5 Department of Pathology, College of Veterinary Medicine, University of Georgia, Athens, Georgia, United States of America

\* rplemper@gsu.edu

**Data Availability Statement:** All relevant data are within the manuscript and its Supporting Information files.

## Abstract

Influenza outbreaks are associated with substantial morbidity, mortality and economic burden. Next generation antivirals are needed to treat seasonal infections and prepare against zoonotic spillover of avian influenza viruses with pandemic potential. Having previously identified oral efficacy of the nucleoside analog 4'-Fluorouridine (4'-FIU, EIDD-2749) against SARS-CoV-2 and respiratory syncytial virus (RSV), we explored activity of the compound against seasonal and highly pathogenic influenza (HPAI) viruses in cell culture, human airway epithelium (HAE) models, and/or two animal models, ferrets and mice, that assess IAV transmission and lethal viral pneumonia, respectively. 4'-FIU inhibited a panel of relevant influenza A and B viruses with nanomolar to sub-micromolar potency in HAE cells. *In vitro* polymerase assays revealed immediate chain termination of IAV polymerase after 4'-FIU incorporation, in contrast to delayed chain termination of SARS-CoV-2 and RSV polymerase. Once-daily oral treatment of ferrets with 2 mg/kg 4'-FIU initiated 12 hours after infection rapidly stopped virus shedding and prevented transmission to untreated sentinels. Treatment of mice infected with a lethal inoculum of pandemic A/CA/07/2009 (H1N1)pdm09 (pdmCa09) with 4'-FIU alleviated pneumonia. Three doses mediated complete survival when treatment was initiated up to 60 hours after infection, indicating a broad time window for effective intervention. Therapeutic oral 4'-FIU ensured survival of animals infected with HPAI A/VN/12/2003 (H5N1) and of immunocompromised mice infected with pdmCa09. Recoverees were protected against homologous reinfection. This study defines the mechanistic foundation for high sensitivity of influenza viruses to 4'-FIU and supports 4'-FIU as developmental candidate for the treatment of seasonal and pandemic influenza.

**Funding:** This study was supported, in part, by public health service grant AI141222 (to R.K.P.) from the NIH/NIAID. The funders had no role in study design, data collection and analysis, decision to publish, or preparation of the manuscript.

**Competing interests:** I have read the journal's policy and the authors of this manuscript have the following competing interests: MGN and GRP are co-inventors on patent WO 2019/1736002 covering composition of matter and use of 4'-FIU (EIDD-2749) and its analogs as an antiviral treatment. This study could affect their personal financial status. RKP reports contract testing from Enanta Pharmaceuticals and Atea Pharmaceuticals, and research support from Gilead Sciences, outside of the described work. All other authors declare that they have no competing interests.

## Author summary

Next-generation antiviral therapeutics are needed to better mitigate seasonal influenza and prepare against zoonotic virus spillover from animal reservoirs. At greatest risk are the immunocompromised and patients infected with highly pathogenic influenza viruses. In this study, we have demonstrated efficacy of a broad-spectrum nucleoside analog, 4'-fluorouridine, against a representative panel of influenza viruses in cell culture, human airway epithelium (HAE) cells, and two animal models, ferrets and mice. Acting as an immediate chain terminator of the influenza virus polymerase, once-daily oral treatment protected against lethal infection with seasonal and highly pathogenic avian influenza viruses, prevented viral transmission to untreated sentinels, and mitigated lethal infection of immunocompromised hosts. These results support the developmental potential of 4'-fluorouridine for treatment of vulnerable patient groups and mitigation of pandemic influenza, providing a path towards a much-needed additional therapeutic option for improved disease management.

## Introduction

Influenza A viruses (IAV) are zoonotic negative sense, segmented RNA viruses with high pandemic potential [1]. Wild aquatic birds are a major natural reservoir for avian influenza viruses, creating high opportunity for spread into the poultry industry and spillover of HPAI viruses into the human population with potentially catastrophic consequences [2,3]. The 2021–2022 H5N1 HPAID epidemic specifically is the largest on record, having reached an unprecedented geographical expansion and requiring the culling of 48 million birds so far [4]. Approximately 14 IAV pandemics have occurred in the past 500 years [5], six of which were concentrated in the last 120 years alone. Most deadly of these outbreaks was the 1918 "Spanish Flu" pandemic, which was caused by an H1N1 subtype IAV of unusual virulence that accounted for approximately 500 million infected people, equating to nearly a third of the world population at the time, and 50 million death [6]. Case fatality rates were particularly high in young children, 20–40 year old adults, and the elderly above 65 years of age [6]. Subsequent pandemics in the 20[th] century included the Asian influenza of 1957 and the Hong Kong influenza of 1968, which were caused by IAV of H2N2 and H3N2 subtypes, respectively [7], and the swine origin H1N1 outbreak of 2009 [8]. Although considered less pathogenic than several of the earlier pandemic strains, pdm09 IAV is still estimated to have caused 151,000 to 575,400 deaths worldwide and major economic damage [9].

 Influenza viruses cause rapid-onset disease with high fever, cough, body aches, rhinorrhea, and nasal congestion [10]. Advance to severe disease is marked by virus invasion of the small airways and viral pneumonia, which can lead to sepsis due to a hyperinflammatory systemic response [11]. Other serious complications include myocarditis, encephalitis, myositis, and multi-organ failure [12,13]. At greatest risk of life-threatening influenza are children less than 2 years of age, older adults, pregnant women, and patients with pre-existing conditions such as asthma, diabetes, or chronic heart disease [14]. Vaccine prophylaxis is available, but the efficacy of the seasonal tetravalent influenza vaccine is moderate, ranging from 40 to 60% even under optimal conditions in interpandemic years [15]. Effectiveness of the vaccine can be substantially lower in older adults, who are at greatest risk of progressing to severe disease, or when vaccine components and circulating strains are poorly matched [15]. Multiple avenues are being pursued to develop a broad-spectrum influenza vaccine, but none have yet achieved regulatory approval [16].

To improve disease management especially in vulnerable patient groups, rapidly respond to endemic outbreaks of seasonal influenza when the vaccine is not optimally matched, and to prepare for the pandemic threat originating from zoonotic transmission of HPAI viruses, effective therapeutics are urgently needed. Three classes of antiviral drugs have received approval by the FDA for treatment of influenza thus far. The M2 channel blocking adamantanes (amantadine and rimantadine), the neuraminidase inhibitors (oral oseltamivir, inhaled zanamivir, intravenous peramivir), and, most recently, the PA endonuclease inhibitor baloxavir marboxil [17]. Of these, use of the adamantanes has been discontinued due to widespread pre-existing resistance in human influenza viruses and natural IAV reservoirs [18]. Pre-existing signature resistance mutations against the neuraminidase inhibitors were furthermore detected in 2009 pandemic viruses [19], giving rise to the concern that future effectiveness of this inhibitor class may be questionable [20]. Although baloxavir marboxil is the newest influenza drug, resistance has developed rapidly and was even observed in human patients during the early clinical trials, resulting in rebound of virus replication [21–23]. Novel therapeutic options are therefore needed to broaden the available anti-influenza virus drug arsenal.

In search of suitable broad-spectrum developmental candidates, we explored in this study antiviral potency, *in vivo* efficacy, and mechanism of action of 4'-FlU against the influenza virus indication. Recently, we reported that 4'-FlU is a highly potent inhibitor of SARS-CoV-2 and pathogens of the mononegavirus order [24,25], establishing oral efficacy in animal models of SARS-CoV-2 and RSV infection. Having demonstrated that this nucleoside analog functions as an immediate chain terminator of influenza virus polymerase, we assessed oral efficacy in two different animal models, mice and ferrets, against pandemic human and HPAI viruses. Influenza treatment paradigms developed in these models outline an unusually broad time window for successful therapeutic interference with a lethal viral challenge. These results support the development of 4'-FlU as a therapeutic candidate for the treatment of influenza with the potential to improve preparedness against zoonotic spillover of avian IAVs with pandemic potential into the human population.

## Results

To explore whether the indication spectrum of 4'-FlU (Fig 1A) extends to the orthomyxoviruses, we examined activity of the compound against seasonal and pandemic IAVs representing H1N1 and H3N2 subtypes and influenza B virus (IBV) isolates representing the Yamagata and Victoria lineages in cell culture (Fig 1B). In parallel, we determined activity of 4'-FlU against a recombinant pandemic A/California/07/2009 (H1N1) (pdmCa09), A/swine/Spain/53207/2004 (H1N1), B/Memphis/20/1996, and B/Malaysia/2506/2004 on non-differentiated primary human airway epithelium (HAE) cells (Fig 1C), explored 4'-FlU potency against a recombinant A/WSN/33 (H1N1)-based reporter virus [26] on a panel of non-differentiated primary HAEs representing 10 distinct donors (S1A Fig), and generated an airway epithelium model through differentiation of HAEs at air-liquid interface to test inhibition of pdmCa09 through basolateral 4'-FlU in a disease-relevant *ex vivo* tissue model (S1B Fig). Most viral isolates tested were efficiently inhibited by the compound, consistently returning 50% and 90% inhibitory concentrations (EC$_{50}$ and EC$_{90}$, respectively) in the nanomolar range with steep Hill slopes independent of host cell type. The least sensitive were A/swine/Spain/53207/2004 (H1N1), B/Memphis/20/1996, and B/Malaysia/2506/2004, but 4'-FlU inhibitory concentrations remained in the low-micromolar range. Recapitulating our previous experience with 4'-FlU performance on respiratory epithelium cultures [24], potency of 4'-FlU against pdmCa09 was greater in the differentiated HAE model (EC$_{90}$ 0.07 μM) than on cell lines (EC$_{90}$ 0.7 μM) or undifferentiated primary cells (EC$_{90}$ 3.33 μM). Combined with a 50% cytotoxic

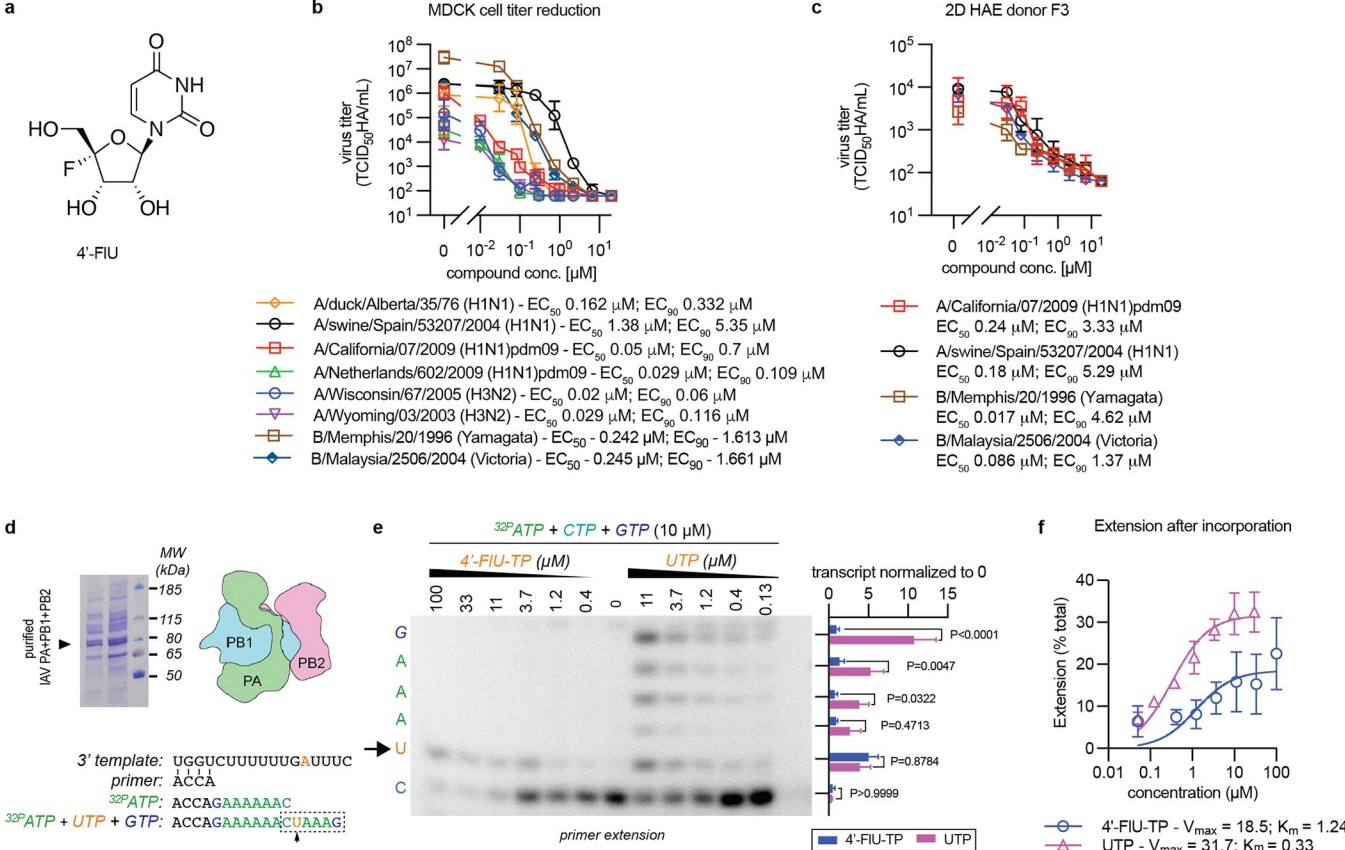

**Fig 1. *In vitro* potency and MOA of 4'-FlU against influenza viruses. a)** Structure of 4'-FlU. **b)** Dose-response assay of 4'-FlU on MDCK cells. Cells were infected with different subtype IAVs or different IBV isolates. **c)** 4'-FlU antiviral activity in 2D HAE cultures. Dose-response assays of 4'-FlU against A/California/07/2009 (H1N1), A/swine/Spain/53207/2004 (H1N1), A/Wisconsin/67/2005 (H3N2), B/Memphis/20/1996 (Yamagata lineage), and B/Malaysia/2506/2004 (Victoria lineage) on undifferentiated HAEs derived from a healthy female donor. Lines in (b-c) intersect, and symbols show, geometric means ± SD (n = 3); $EC_{50}$ and $EC_{90}$ values based on 4-parameter variable slope regression modeling are given. **d)** Coomassie blue staining of purified recombinant IAV RdRP proteins after gel electrophoresis and RNA template sequences used in primer extension assays. **e)** *In vitro* RdRP assay in the presence of $^{32}$P-ATP, CTP, GTP, and UTP or 4'-FlU-TP as indicated. Representative autoradiogram showing the sequence section highlighted by dashed box in (d); the first incorporation of UTP or 4'-FlU-TP is after position *i* = 8 of the amplicon (arrow). The sequence of the amplicon and results of phosphoimager-quantitation of relative signal intensities observed in the presence of 11 μM UTP or 4'-FlU-TP are specified to the left and right of the autoradiogram, respectively. Quantitation graph shows mean values of independent experiments (n = 3) ± SD; analysis with 1-way ANOVA with Dunnett's *post hoc* test; P values are shown in the graph. Uncropped autoradiogram and replicates are provided in (S2 Fig). **F)** Kinetic analysis of 4'-FlU-TP and UTP incorporation into the amplicon shown in (e) and (S2 Fig). Lines represent non-linear regression kinetics with Michaelis-Menten model, $K_m$ and $V_{max}$ are shown, error bars represent 95% CI.

concentration ($CC_{50}$) of 4'-FlU of 468 μM [24] in primary HAEs from the same donor as used in the differentiated HAE model, we calculated a selectivity index (SI = $CC_{50}/EC_{50}$) of 4'-FlU against pdmCa09 of 15,600 in this disease-relevant human tissue model. Transepithelial electrical resistance (TEER) between basolateral and apical chamber of the air-liquid interface cultures was unchanged throughout the study (S1C Fig), confirming our previous observation that 4'-FlU does not disrupt epithelium integrity [24] and, in the case of vehicle-treated wells, that the epithelium maintains barrier function despite productive IAV replication [27].

## 4'-FlU is an immediate chain terminator of influenza virus polymerase

Through biochemical RdRP assays using purified recombinant RSV and SARS-CoV-2 polymerase complexes and synthetic template RNA, we have demonstrated that incorporation of 4'-FlU triphosphate (4'-FlU-TP) into nascent chain RNA triggers delayed termination of the

RNA-dependent RNA polymerases from a negative sense (RSV) and a positive sense (SARS-CoV-2) RNA virus. Whereas variations in stalling efficiencies were observed between sequences and polymerase, position $i+3$ marked the predominant stalling site [24]. Following expression of IAV RdRP PA, PB1, and PB2 subunits in insect cells, we subjected affinity chromatography purified, reconstituted IAV polymerase to the equivalent biochemical RdRP assay in the presence of increasing concentrations of 4'-FlU-TP or UTP (Figs 1D, 1E and S2A to S2E). 4'-FlU-TP was incorporated instead of UTP when the polymerase reached the first adenosine in the template RNA as we had noted before in RSV and SARS-CoV-2 RdRP assays [24], but then triggered immediate chain termination of IAV RdRP. Densitometric quantitation of relative amplicon intensities to compare incorporation rates of the analog relative to UTP revealed an approximately 4-fold higher affinity of IAV polymerase for endogenous UTP than for 4'-FlU-TP (Fig 1F).

These results demonstrate high sensitivity of influenza virus polymerases to 4'-FlU and identify sequence-independent immediate chain termination at position $i$ as a predominant mechanism of influenza virus RdRP inhibition by the compound.

## Pharmacokinetic (PK) properties of 4'-FlU in mouse plasma and tissues

We have previously demonstrated oral bioavailability of 4'-FlU in ferrets and efficient intracellular anabolism to 4'-FlU-TP with prolonged tissue exposure levels [24]. To assess cross-species conservation of PK profiles between IAV efficacy models used in this study, ferrets and mice, we subjected the compound to a single-dose oral PK study in mice, measuring 4'-FlU plasma concentrations and corresponding 5'-triphosphate levels in selected soft tissues after oral dosing at 1.5 mg/kg body weight (Fig 2A and 2B). Free 4'-FlU reached a maximal plasma exposure of approximately 1 μM 90 minutes after oral administration, far exceeding the cell culture $EC_{90}$ concentration against the influenza virus test panel. Plasma levels were sustained high over a 12-hour period after dosing. Corresponding 4'-FlU-TP tissue exposure levels 12 hours after dosing were approximately 1 nmol/g tissue in all organs but brain and kidney, underscoring suitability of 4'-FlU for *in vivo* efficacy testing against IAV.

## Late-onset treatment with 4'-FlU mediates complete survival of IAV-infected animals

For mouse efficacy studies, animals were infected intranasally with a lethal inoculum amount of pdmCa09, which replicates efficiently in mice without a requirement for species adaptation [28]. Infection results in severe viral pneumonia with major clinical signs such as weight loss and hypothermia. To establish the lowest efficacious dose, we examined 3 4'-FlU dose levels in a survival study, each administered by oral gavage on three consecutive days in a once daily (q.d.) regimen starting at the time of infection (Fig 2C). All animals treated at the 2 mg/kg dose level survived, whereas all mice of the vehicle-treated and lowest dose (0.08 mg/kg) groups succumbed to infection within 5 days of inoculation (Fig 2D) and experienced clinical signs (S3A and S3B Fig). Animals receiving the intermediate 4'-FlU dose (0.4 mg/kg) had prolonged survival, but ultimately all died.

Having established 2 mg/kg 4'-FlU administered q.d. as a low efficacious dose, we next determined the latest time for initiation of effective treatment at this dose level (Fig 2E). All vehicle-treated animals again succumbed to infection within 5 days of inoculation (Fig 2F). In contrast, treatment start at 2.5 days after infection, which corresponds to 0.5 days after the onset of major clinical signs (S3C and S3D Fig) ensured complete survival, and 80% of animals of a group receiving the first dose 3 days after infection survived, outlining a broad therapeutic time window for treatment of influenza with 4'-FlU. Therapeutic benefit of late-onset

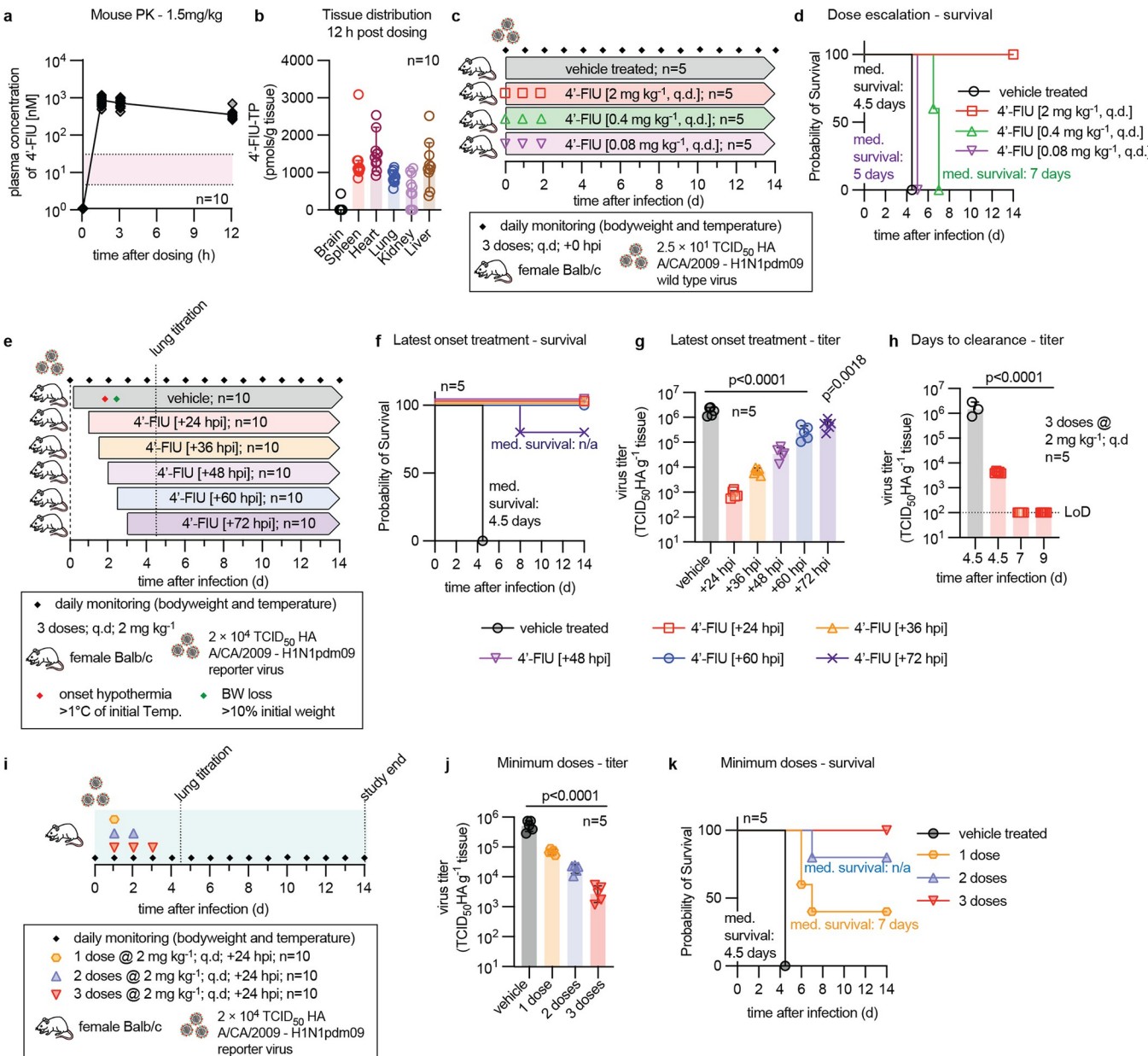

**Fig 2. Treatment paradigms of 4'-FlU in mice. a)** Mouse plasma exposure after a single oral dose of 1.5 mg/kg bodyweight. The red area denotes cell culture $EC_{90} \pm 1 \times SD$ against WSN, all donors as shown in (S1A Fig). **b)** Tissue distribution of bioactive anabolite 4'-FlU-TP in animals shown in (a) 12 hours after dosing. **c)** Schematic of dose-to-failure study against standard recA/CA/2009 (H1N1). **d)** Survival of animals treated as specified in (c). Median survival (med. survival) time in days is specified; Kaplan-Meier simple survival analysis. **e)** Study schematic to determine the therapeutic time window of oral 4'-FlU in mice, using recA/CA/2009-maxGFP-HA (H1N1) as viral target. **f)** Survival of animals treated as specified in (e). Median survival time in days is specified; Kaplan-Meier simple survival analysis. **g)** Lung viral load of a set of animals treated as in (e) was determined 4.5 days after infection. **h)** Time-to-viral-clearance study. Mice were infected and treated with 4'-FlU at 2 mg/kg starting 24 hours after infection and continued q.d. Lung virus load was determined in vehicle-treated animals 4.5 days after infection, and in 4'-FlU-treated animals 4.5, 7, and 9 days after infection. **i)** Schematic of minimal-number-of-doses finding study. **j)** Lung viral load of animals treated as in (i), determined 4.5 days after infection. **k)** Survival of a set of animals treated as in (i). Median survival time in days is specified; Kaplan-Meier simple survival analysis. Columns in (b) represent data medians with 95% CI, columns in (g-h,j) represent geometric means ± SD; symbols specify individual animals; statistical analysis in (g-h,j) with 1-way ANOVA with Dunnett's *post hoc* test.

treatment was corroborated by a significant reduction in lung virus load compared to vehicle-treated controls at 4.5 days after infection, determined in a parallel set of equally treated animals (Fig 2G). Within 7 days, virus became undetectable in lungs of animals that were treated starting 24 hours after infection with oral 4'-FlU (2 mg/kg q.d.; 3 doses total), indicating that viral replication had ceased (Fig 2H). No rebound of virus replication was observed within a 9-day period after infection (S3E and S3F Fig). These results define a broad therapeutic time window for effective treatment of IAV infection with 4'-FlU.

To determine the minimal number of oral 4'-FlU doses (2 mg/kg q.d.) for complete survival, we again initiated treatment 24 hours after infection, in this study comparing the effect of a single vs double or triple doses (Fig 2I). Lung virus load was assessed 4.5 days after infection and survival monitored in a parallel set of equally treated animals. The number of doses administered correlated with antiviral effect size, but even a single dose of 4'-FlU resulted in a statistically significant reduction in lung virus load of approximately one order of magnitude (Fig 2J). Reduced viral burden translated to survival of some animals in the single and double dose groups. However, three doses were required for full therapeutic benefit (Fig 2K) and only mild clinical signs (S3G and S3H Fig).

## 4'-FlU effect on IAV transmission in the ferret model

Since mice, in contrast to ferrets, do not shed IAVs, we next explored anti-IAV efficacy of 4'-FlU in the ferret model. Guided by the dosing regimen established in mice, we treated animals infected intranasally with pdmCa09 with either 2 mg/kg (standard dose) or 0.5 mg/kg (low dose), administered orally starting 24 hours after infection (Fig 3A). Animals received one (2 mg/kg level) or two doses (either 2 or 0.5 mg/kg level) total. Fever is the predominant clinical sign associated with pdmCa09 IAV infection in ferrets. Monitoring body temperature continuously telemetrically, we observed significant reduction in all treated animals within 24 hours of the first dose, whereas vehicle-treated animals showed a prolonged fever plateau (S4A Fig). Viral titers in nasal lavages were statistically significantly reduced compared to those of vehicle animals, but effect size was lower in animals of the low dose than the standard dose group, and virus rebounded 2.5 days after infection in the single standard dose group (Fig 3B). In contrast, two standard doses 4'-FlU given q.d. were sterilizing within 2 days of treatment initiation. Virus load in nasal turbinates assessed 4 days after infection corroborated the lavage titers, revealing a dose-dependent significant antiviral effect of two doses, whereas a single dose had no significant effect on virus titer in turbinates at the time of tissue harvest (Fig 3C). Examination of lower respiratory tract tissues, trachea, and bronchoalveolar lavage fluid (BALF) revealed, however, that a single standard dose of 4'-FlU administered at the time of onset of clinical signs completely suppressed the development of viral pneumonia (Fig 3D).

To determine the effect of 4'-FlU on virus spread, we treated in a direct-contact transmission study source ferrets infected with pdmCa09 with the compound as before, starting 12 or 24 hours after infection (Fig 3E). Co-housing of source ferrets with untreated sentinels in a 1:1 ratio was initiated 2.5 days after study start. pdmCa09 rapidly spread from vehicle-treated source ferrets to their direct contacts, which first became virus-positive in nasal lavages only 12 hours after the beginning of co-housing (Fig 3F) and had progressed to viral pneumonia with high turbinate, lung and trachea virus load by study end (Fig 3G and 3H) and clinical signs (S4B to S4E Fig). Treatment of source animals with 4'-FlU initiated 12 or 24 hours after infection in each case resulted in exponential decline in shed virus load, but virus was undetectable in lavages of the 12-hour treatment group by the time of co-housing whereas approximately $10^4$ TCID$_{50}$ units/ml lavage remained detectable in lavages of the 24-hour treatment group at that time. We did not detect any virus transmission to untreated sentinels of the

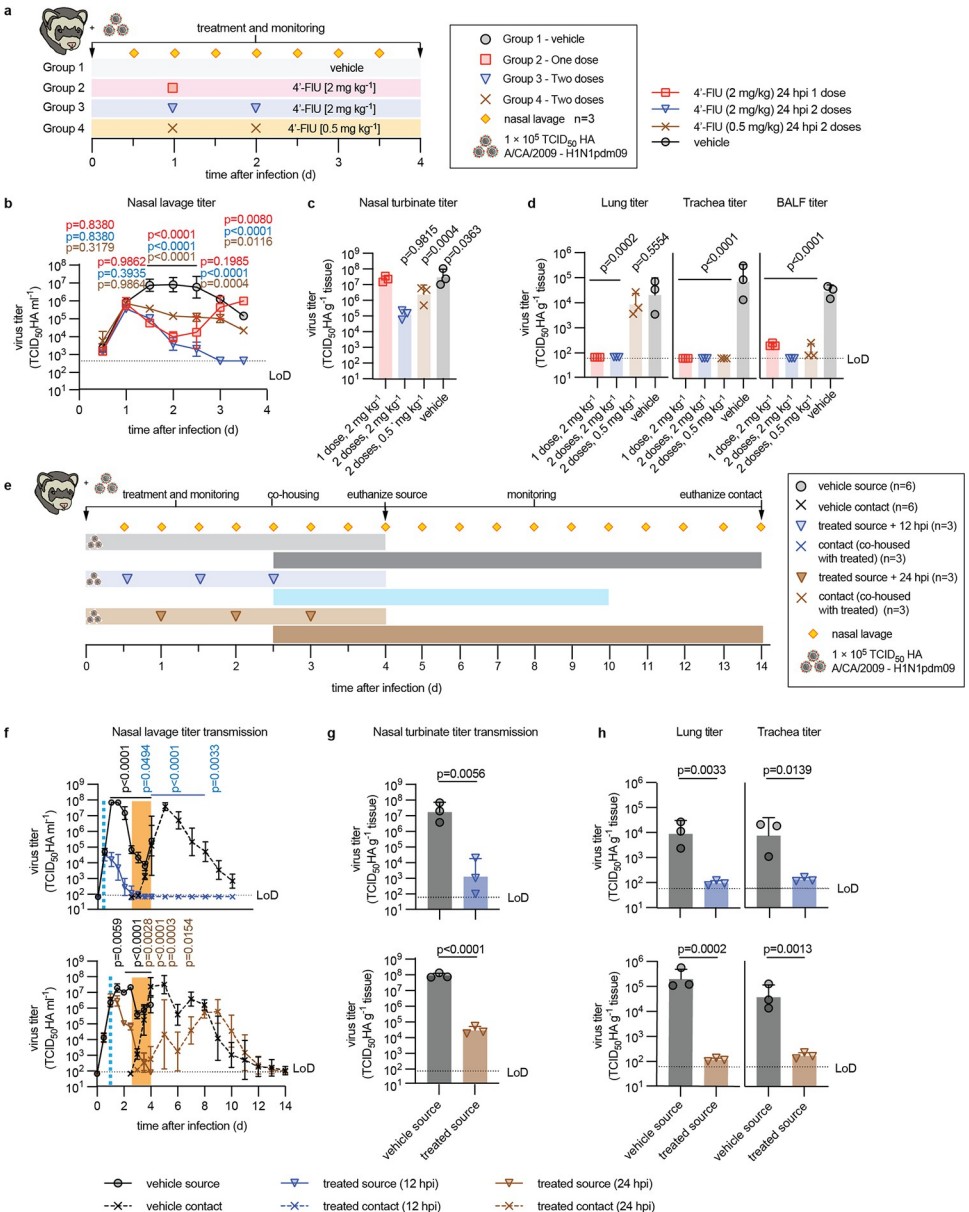

**Fig 3.** **In vivo efficacy of 4'-FlU in the ferret transmission model. a)** Ferret efficacy study schematic. **b)** Ferret shed viral titers in nasal lavages (n = 3). **c-d)** Tissue and BALF viral titers determined 4 days after infection. **e)** Transmission study schematic. Treatment of source ferrets was started 12 or 24 hours after infection, animals were co-housed with untreated contact ferrets 2.5 days after infection. **f)** Ferret shed viral titers in nasal lavages of source animals and their sentinels. Light blue dashed line marks time of treatment onset; orange box highlights co-housing period. **g-h)** Virus load in nasal turbinates (g) and lung and trachea tissues (h) in source animals, collected 4 days after infection, respectively. Lines in (b,f) intersect, and symbols show, geometric means ± SD; 2-way ANOVA with Tukey's *post hoc* test, P values are specified (in (f), P values in black compare source animals, P values in blue and brown compare contact ferrets). Columns in (c-d,g-h) represent geometric means ± SD; symbols show individual animals; 1-way ANOVA with Dunnett's *post hoc* test (c-d) or unpaired t-test (g-h); P values are specified.

12-hour group. However, low-grade transmission to direct contacts of the 24-hour treatment group occurred, characterized by delayed onset of virus shedding compared to sentinels of vehicle-treated source animals and a reduced peak shed virus load (Fig 3H). Shedding from all sentinels that became virus-positive ceased approximately 12 days after the beginning of co-

housing. These results confirm cross-species anti-IAV efficacy of 4'-FlU, reveal that a single oral dose is sufficient to prevent advance to severe influenza, and demonstrate a shortened window in which a treated case remains fully infectious.

## Mitigation of IAV pathogenesis and lung histopathology by 4'-FlU

To better understand the impact of 4'-FlU on mitigating immunopathogenesis, the major driver of clinical signs associated with influenza virus infection, we determined inflammatory cytokine profiles for a 10-day period after infection of mice with pdmCa09 using a 6-plex Th17 cytokine assay (Fig 4A). Treated animals received the 3-dose standard 4'-FlU regimen as before, started 24 hours after infection. Lung tissue for histopathology was extracted from parallel sets of equally treated animals 5 or 21 days after infection. Lung titers in vehicle-treated animals demonstrated fast-onset viral pneumonia, approaching a burden of $10^6$ infectious units/g lung tissue 4 days after infection, when all animals of the vehicle group had succumbed to infection with major clinical signs (Figs 4B, S5A and S5B). Pro-inflammatory cytokines rapidly increased in the BALF of these animals, revealing a very strong antiviral response until death of the vehicle animals as expected after infection with pdmCa09 (Figs 4C, 4D and S6). Treatment with 4'-FlU again efficiently suppressed virus replication within 24 hours of the first dose (Fig 4B). Coinciding with the reduction in virus burden of treated animals by approximately two orders of magnitude by day 3 after infection, the proinflammatory response was greatly alleviated (Figs 4C, 4D and S6) and all treated animals survived (S5A Fig).

Consistent with a better clinical outcome (S7A and S7B Fig), lowered viral load, and mitigated innate antiviral response in 4'-FlU treated vehicle animals, analysis of lung histopathology 5 days after infection revealed reduced lung lesions and alleviated cellular infiltrates in lung tissue of treated animals (Figs 4E and S8). Overall combined histopathology scores of individual animals representing the degree of cellular infiltrates into alveoli and bronchioles, the extent of perivascular cuffing, and the severity of interstitial pneumonia, pleuritis, and vasculitis were statistically significantly lower in the treatment than vehicle group (Figs 4F and S9).

## 4'-FlU control of influenza in immunocompromised hosts and of HPAI infection

In the clinic, the risk of progression to severe disease and poor overall outcome is greatly increased when immunocompetence of the patient is impaired or after zoonotic spillover of HPAIs into the human population [29,30]. To first assess the potential of 4'-FlU to pharmacologically manage influenza in high-risk hosts, we infected interferon α/β receptor knockout (IFNAR1 KO) and V(D)J recombination activation gene RAG-1 KO mice with pdmCa09. Whereas the former cannot mount an effective type I IFN innate response [31,32], the latter are compromised in their cell-mediated adaptive immunity due to a lack of mature T and B cells [33]. Oral treatment with 4'-FlU was initiated 24 hours after infection at standard and a high-dose (10 mg/kg) level, then continued q.d. for a total of 7 doses of each group (Fig 5A). Independent of 4'-FlU dose level, all treated animals of the RAG-1 KO and IFNAR1 KO groups survived, whereas all animals in the vehicle-treated groups developed severe clinical signs (S10A to S10D Fig) and succumbed to infection within 4 days (Fig 5B). Reduction in lung virus load of treated vs vehicle animals after three q.d. 4'-FlU doses (day 4 after infection) was dose level-dependent, but even at the standard dose, this reduction exceeded two orders of magnitude in either knockout mouse strain (Fig 5C).

Next, we explored 4'-FlU efficacy against HPAIs, initially testing potency against polymerases derived from A/CA/07/2009 (H1N1), A/VN/12/2003 (H5N1), and A/Anhui/1/2013 (H7N9) in minireplicon dose-response assays (Fig 5D). The compound dose-dependently

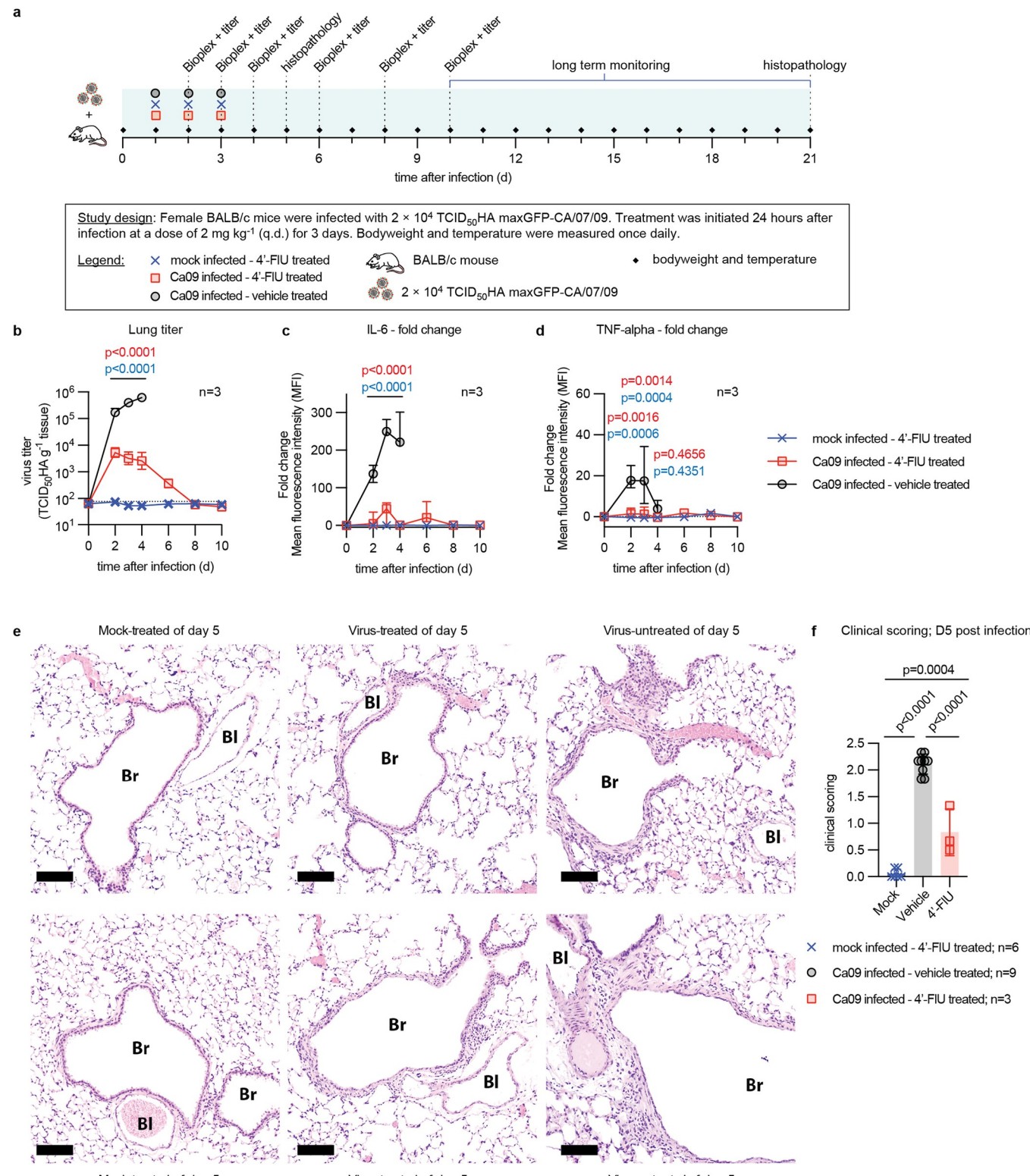

**Fig 4. Effect of 4'-FlU on the antiviral immune response and lung histopathology. a)** *In vivo* Bioplex and histopathology study schematic. **b)** Lung viral titers in animals treated as in (a). **c-d)** Changes in IL-6 (c) and TNF-α (d) levels present in BALF of animals treated as in (a), relative to levels at time of infection. Lines in (b) intersect, and symbols show, geometric means ± SD, lines in (c-d) intersect, and symbols show, data medians with 95% CI; 2-way ANOVA with Tukey's *post hoc* test; P values are given. **e)** Representative photomicrographs of lung tissue extracted 5 days after infection of animals treated as in (a). Tissues

of two individual animals per study arm are shown at 10× magnification; scale bar denotes 100 μm; Br, Bronchiole; Bl, Blood vessel. **f)** Histopathology scores of animals treated as in (a). Lungs were extracted 5 days after infection. Scores for each animal represents a mean of individual alveolitis, bronchiolitis, vasculitis, pleuritis, perivascular cuffing (PVC), and interstitial pneumonia (IP) scores. Columns represent data medians with 95% CI; symbols show mean scores for each individual animal; 1-way ANOVA with Dunnett's *post hoc* test, P values and n values for each study arm are specified.

inhibited activity of either HPAI-derived polymerase complex, reaching active concentrations resembling that observed against pdmCa09 RdRP. Selecting A/VN/12/2003 (H5N1), which causes lethal encephalitis in mice [34], for a proof-of-concept efficacy study of 4'-FlU against zoonotic HPAIs, we started treatment at the 5 mg/kg dose level 12 or 24 hours after infection and continued q.d. thereafter (Fig 5E). This elevated dose was chosen to compensate for lower brain tissue exposure of 4'-FlU (Fig 2A) when treating infection by a CNS-invasive IAV. None of the animals of either 4'-FlU treatment group developed clinical signs even when treatment was initiated 24 hours after infection (Fig 5F), and all treated animals survived (Fig 5G). In contrast, 19 out of 20 vehicle-treated animals succumbed to A/VN/12/2003 (H5N1) with a median survival time of 9 days after infection, developing clinical signs on day 7 after infection followed by rapid further deterioration (Fig 5F). Vehicle-treated animals experienced high lung virus loads three days after infection (Fig 5H). Treatment reduced HPAI burden by several orders of magnitude to limit of detection. These results confirm uncompromised efficacy of 4'-FlU in high-risk immunocompromised hosts and against highly pathogenic zoonotic viruses with high pandemic concern.

## Anti-IAV immune status of 4'-FlU-treated recoveries

Highly efficacious antivirals may suppress host immune activation, especially when treatment is started early in the course of infection, which may leave the host susceptible to subsequent re-infection and may potentially even result in exacerbated disease [35–37]. To explore the effect of 4'-FlU on subsequent homotypic IAV rechallenge of recoveries, we infected animals with pdmCa09 and started standard 3-dose 4'-FlU treatment at three time points, therapeutically at 24 and 48 hours after infection and prophylactically at 12 hours before infection (Fig 6A). For this study, mice were inoculated with aerosolized virus in a temperature and airflow-controlled aerosol chamber using an Aeroneb nebulizer (S11A and S11B Fig) to better mimic natural infection. Prior to study start, we performed pilot experiments to determine suitable exposure time (S12A to S12C Fig), infectious dose (S13A and S13B Fig) and proof-of-concept for 4'-FlU efficacy (S14A to S14D Fig) after infection with aerosolized virus. Mirroring the outcome of our previous efficacy studies, animals of all treatment groups survived the original infection, whereas all animals of the vehicle group reached predefined clinical endpoints 9 days after infection and had to be terminated (Fig 6B) with major clinical signs (S15A and S15B Fig). To assess the status of humoral anti-Ca09 immunity, we collected blood samples from the treated recoverees 4, 8, and 12 weeks after the original infection and determined neutralizing antibody (nAb) titers. Mice in both therapeutic treatment groups mounted a robust humoral anti-IAV response, whereas no nAbs were detectable in the prophylactically treated animals 8 weeks after the original infection (Fig 6C). Upon homotypic rechallenge with aerosolized pdmCa09 10 weeks after the original infection, however, all 4'-FlU-experienced recoverees had a significant benefit over a new set of IAV-naïve mice (Fig 6D). All animals of the therapeutic groups survived with significant boost in nAb titers relative to week 8 levels. Remarkably, four of the five animals treated prophylactically with 4'-FlU at the time of the original infection also survived the homotypic rechallenge and mounted a robust humoral anti-Ca09 response by week 12. In contrast, all newly added IAV-inexperienced animals succumbed to infection within 9 days, confirming efficient virus delivery through aerosolization

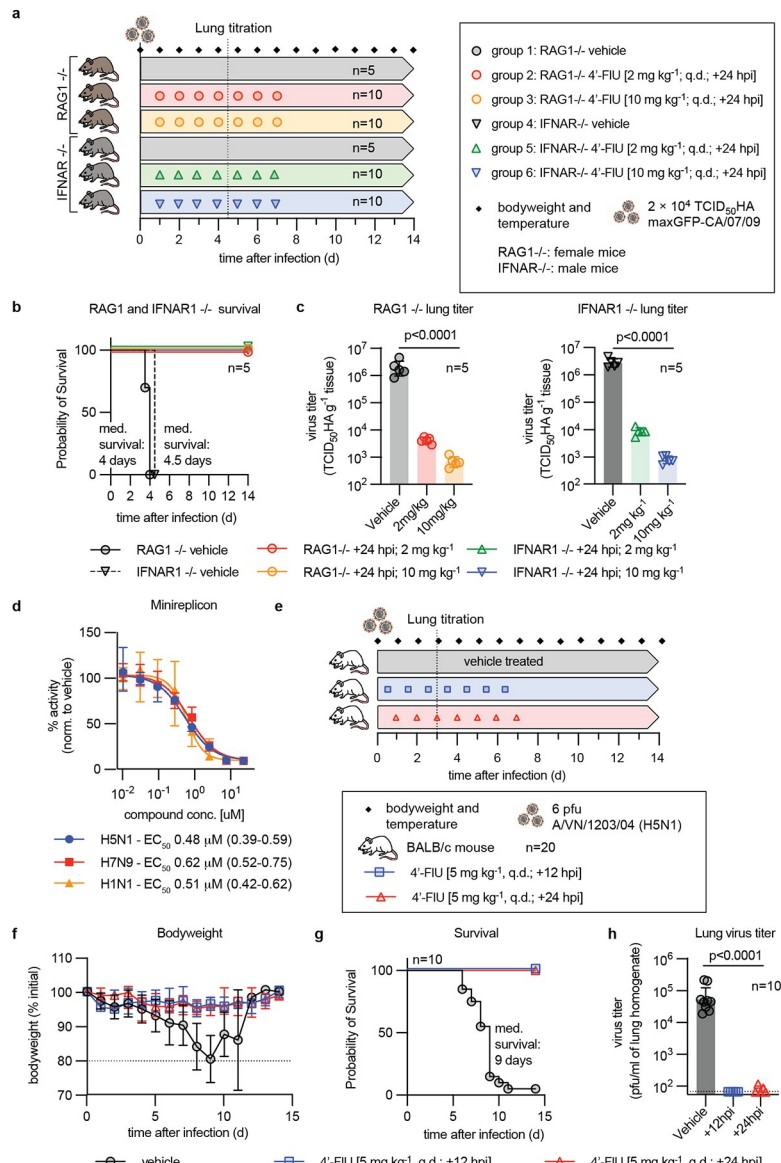

**Fig 5. Efficacy of 4'-FlU in an immunocompromised host and against HPAI. a)** Efficacy study schematic in immunocompromised mice, lacking B and T cells (RAG1 KO) or IFN1 receptor function (IFNar1 KO). **b)** Survival study of animals treated as shown in (a). Median survival time in days is specified; Kaplan-Meier simple survival analysis. **c)** Lung viral load in animals from (a), determined 4.5 days after infection. **d)** Dose-response minigenome assay with RdRP complexes derived from IAV subtypes H1N1, H5N1, and H7N9. Lines represent 4-parameter variable slope regression models; symbols show data medians with 95% CI; n = 3. $EC_{50}$ concentrations and 95% CI are specified. **e)** Efficacy study schematic of 4'-FlU against HPAI H5N1. **f)** Body weight measurements of animals shown in (e). Lines intersect, and symbols show, data means ± SD, normalized to animal body weight at study start. Dashed line shows predefined endpoint. **g)** Survival study of animals infected with HPAI and treated as shown in (e). Median survival time in days is specified; Kaplan-Meier simple survival analysis. **h)** Lung virus load on day 3 after infection. Columns in (c,h) represent geometric means ± SD; symbols show individual animals; 1-way ANOVA with Dunnett's *post hoc* test; P values and n values are specified.

at rechallenge. Animals of the treatment group remained disease-free during rechallenge, and clinical signs were alleviated in mice of the prophylactic group (S15C and S15D Fig). Independent of the original 4'-FlU regimen applied, we did not detect any signs of exacerbated disease upon rechallenge of treatment-experienced recoverees.

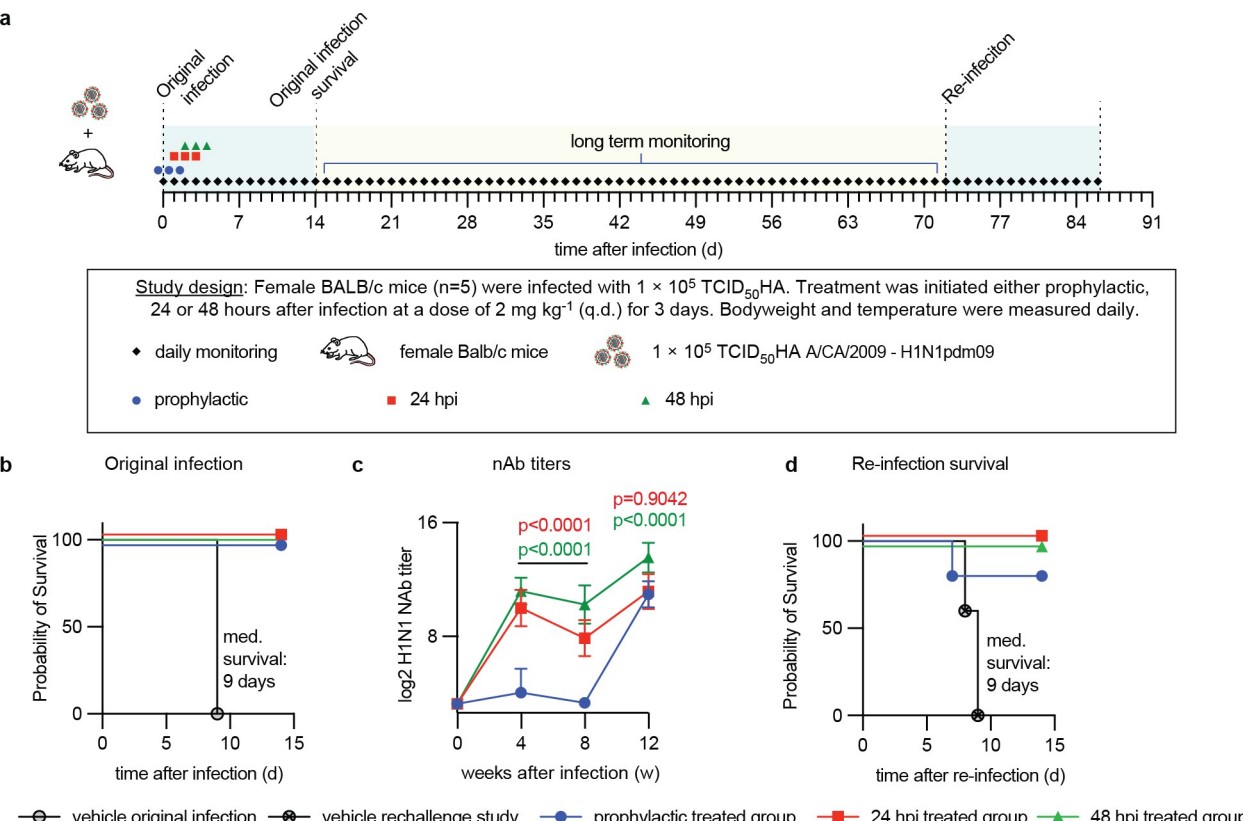

**Fig 6. Reinfection of 4'-FlU-experienced animals with homotypic H1N1. a)** Schematic of the treatment and reinfection study. For infection and reinfection, mice received aerosolized pdmCa09. **b)** Survival study of animals infected, treated, and reinfected as shown in (a). Median survival time in days is specified; Kaplan-Meier simple survival analysis. **c)** Anti-H1N1 neutralizing antibody (nAb) titers developing in animals from (a). Lines intersect, and symbols show, geometric means ± SD; 2-way ANOVA with Tukey's *post hoc* test; P values are specified. **d)** Survival of animals after re-infection. Median survival time in days is specified; Kaplan-Meier simple survival analysis. N numbers for animals in (a-d) are specified in (a).

## Discussion

Treatment of influenza has remained challenging, despite the current availability of two different FDA-approved drug classes, the neuraminidase and endonuclease inhibitors [38]. The main obstacles to more successful antiviral therapies are narrow therapeutic time windows [39], poor patient compliance [40], and impaired immunocompetence of older adults at greatest risk of advance to severe disease [17], defining key developmental objectives for the next generation of influenza therapeutics.

Demonstrating antiviral potency in cell culture, human airway epithelium cells, and/or efficacy in two relevant *in vivo* infection models, ferrets and mice, this study establishes oral efficacy of once-daily administered 4'-FlU, a broad-spectrum antiviral ribonucleoside analog [24,25], against seasonal, pandemic, and HPAI IAV strains. Designed to expand the first-line pharmacological arsenal against RNA viruses of pandemic concern, we have previously shown activity of 4'-FlU against SARS-CoV-2, pneumoviruses such as RSV, paramyxoviruses, and other members of the mononegaviruses [24]. Mechanistic characterization consistently demonstrated that incorporation of 4'-FlU-TP into the nascent RNA strand induces polymerase chain termination [24]. Unique to 4'-FlU of all chain-terminating ribonucleoside analogs analyzed [25], however, the modus of 4'-FlU-induced termination varies dependent on RdRP target. IAV RdRP stalled immediately upon incorporation, RSV polymerase termination was

delayed at position *i*+3 after incorporation, and SARS-CoV-2 polymerase termination was delayed and dependent on sequence context, requiring incorporation of at least two 4'-FlU moieties in close repetition [24]. Delayed chain termination typically reflects that incorporation of the nucleoside analog alters secondary structure of the nascent strand, which interferes with polymerase processivity [25,41,42]. Accordingly, we hypothesize that distinct mechanisms of action (MOAs) against betacoronaviruses, pneumoviruses, and orthomyxoviruses are due to different capabilities of the respective polymerase complexes to accommodate 4'-FlU-specific secondary structure changes.

In clinical trials of baloxavir marboxil, viral resistance emerged rapidly, resulting in rebound of virus replication in 82.1% of treated patients [23]. Whole genome sequencing of 4'-FlU-experienced virus populations isolated from infected and treated animals showed no allele-dominant divergence from the inoculum population [24]. However, future full resistance profiling of 4'-FlU against influenza viruses will be required to better appreciate the height of the genetic barrier against viral escape from inhibition and explore the molecular basis for the slight virus strain-dependent differences in sensitivity to 4'-FlU that we observed in cell culture dose-response assays. Greater potency of 4'-FlU on differentiated HAE models compared undifferentiated HAEs and immortal cell lines especially likely reflects more efficient intracellular anabolism of the compound to the bioactive triphosphate form 4'-FlU-TP and altered viral replication kinetics in differentiated versus undifferentiated HAEs.

Very likely representing a direct consequence of the distinct MOAs, we found influenza viruses to be highly sensitive to inhibition by 4'-FlU, followed by RSV and SARS-CoV-2. Dose-finding in the mouse model identified effective levels in the low mg/kg range and three doses administered q.d. to be sufficient for achieving full therapeutic benefit, which resembled efficacy performance of baloxavir marboxil [43] and far outpaced that of the neuraminidase inhibitors [44]. Whereas therapeutic oseltamivir lacked efficacy in mice infected with pdm09 IAV isolates [45] and baloxavir marboxil ensured only partial survival when treatment was started later than 24 hours after infection [43], treatment with 4'-FlU mediated unprecedented complete survival of all treated animals when the inhibitor was first administered as late as 60 hours after infection, at the time when lung virus burden approached peak titer [46] and major clinical signs had manifested. If predictive of the treatment of human disease, this substantially widened therapeutic time window may be game-changing for influenza therapy.

Since clinical signs of influenza are largely a result of immunopathogenesis [47], we asked whether 4'-FlU may impact the quality of the antiviral immune response and potentially trigger exacerbated disease upon homotypic re-infection. Cytokine profiling showed an alleviated pro-inflammatory response in animals dosed therapeutically with 4'-FlU, but assessment of neutralizing antibody titers revealed that each of the treated animals had mounted a robust humoral response. Consistent with our previous observation of low cytotoxic and cytostatic potential of 4'-FlU [24], these results indicate that mitigated immunopathogenesis in treated animals is a consequence of reduced virus load rather than a direct immunomodulatory effect of the compound. Although essentially sterilizing, even prophylactic treatment resulted in partial protection of animals at re-infection despite the absence of neutralizing antibodies, indicating that treated animals also mounted a viable cell-mediated adaptive response [48,49].

Beyond individual patient benefit through mitigation of lethal infection and alleviation of viral pneumonia and lung immunopathogenesis, 4'-FlU statistically significant reduced upper respiratory tract virus load and viral shedding when treating 12 hours after infection, resulting in fully suppressed spread to untreated direct-contact sentinels in the highly sensitive ferret transmission model [50]. Later treatment initiation still reduced transmission efficiency, manifesting in delayed onset of virus replication and reduced upper respiratory tract peak virus load, which serves as a biomarker for risk of progression to severe viral pneumonia [51]. If

equally applicable to the human host, pharmacological block or mitigation of transmission provides a promising path towards outbreak control through efficient interruption of community transmission chains, mitigating the health threat and economic burden [52] of endemic and pandemic influenza.

At greatest risk of life-threatening disease are the immunocompromised [53,54] and patients infected with zoonotic HPAI viruses [55,56]. When assessed in an immunocompromised mouse model, baloxavir marboxil returned a significantly prolonged time to death, but ultimately all animals succumbed to the infection [57]. In contrast, treatment with 4'-FlU mediated complete survival of all animals in two distinct immunocompromised mouse models, underscoring powerful antiviral activity that does not strictly depend on a fully immunocompetent host to clear the infection [58]. Adding efficient control of highly pathogenic zoonotic IAVs, we propose that 4'-FlU meets key efficacy requirements of a next-generation antiviral clinical candidate that strengthens pandemic preparedness and provides a much-needed alternative option for management of seasonal and pandemic influenza viruses.

## Material and methods

### Ethics statement

All animal work was performed in compliance with the *Guide for the Care and Use of Laboratory Animals* of the National Institutes of Health and the Animal Welfare Act Code of Federal Regulations. Experiments involving mice and ferrets were approved by the Georgia State University Institutional Animal Care and Use Committee (IACUC) under protocols A20012 and A21020, respectively. *In vivo* experimentation with HPAI viruses was approved by the University of Georgia at Athens IACUC under protocol A2020 03–033. All experiments using infectious material were approved by the Georgia State University and the University of Georgia at Athens Institutional Biosafety Committees (IBCs) and performed in BSL-2/ABSL-2 or BSL-3/ABSL-3 containment facilities, respectively. Experiments involving highly pathogenic avian influenza (HPAI) viruses were reviewed and approved by the IBC of the University of Georgia at Athens and were conducted in enhanced biosafety level 3 (BSL3+) containment according to guidelines for the use of select agents approved by the Centers for Disease Control and Prevention (CDC).

### Study design

Cells, ferrets and mice were used as *in vitro*, *ex vivo* and *in vivo* models, respectively, to examine efficacy of 4'-FlU against influenza infections. Biochemical RdRP assays were added for mechanistic characterization against the influenza virus target. Viruses were administered through intranasal or aerosolized (mice only) inoculation and virus load monitored periodically in nasal lavages (ferrets only), and in respiratory tissues of ferrets and mice extracted 4 days and 3–14 days, respectively, after infection. Virus titers were determined through $TCID_{50}$-titration.

### Cells, viruses, and compound synthesis

MDCK (ATCC CCL-34) and 293T (ATCC CRL-3216) cells were cultured in Dulbecco's modified Eagle's medium (DMEM) supplemented with 7.5% heat-inactivated fetal bovine serum (FBS) at 37°C and 5% $CO_2$. Human primary cells were grown in epithelial differentiation medium [24]. A/swine/Spain/53207/2004 (H1N1), A/Wisconsin/67/2005 (H3N2), A/California/07/2009 (H1N1)pdm09, A/Netherlands/602/2009 (H1N1)pdm09, A/WSN/33 (H1N1), A/Wyoming/03/2003 (H3N2), B/Memphis/20/1996 (Yamagata), and B/Malaysia/2506/2004

(Victoria) were propagated on MDCK cells using serum-free DMEM supplemented with 0.5% Trypsin. All influenza A viruses were grown at 37˚C, influenza B viruses at 32˚C. A/Viet Nam/1203/2004 (H5N1) and A/duck/Alberta/35/76 (H1N1) were propagated in the allantoic cavity of embryonated hen eggs at 37˚C for 24 or 48 hours, respectively, clarified, aliquoted, and stored at −80˚C. Virus stock titers were determined through $TCID_{50}$-titration or plaque assay (A/Viet Nam/1203/2004 (H5N1)), stocks were stored in aliquots at -80˚C. Recombinant influenza viruses were recovered using an 8-plasmid system generated for A/CA/07/2009 (H1N1) that was based on original reports for laboratory-adapted IAV [59]. Reporter viruses were generated through insertion of a nanoLuc or maxGFP-encoding ORF in frame at the 5'-end of the viral HA ORF, separated by a 2A cleavage site as described for other recombinant IAVs [60]. Genetic stability of the resulting nanoLuc-HA and maxGFP-HA segments was validated through sequence confirmation after 10 consecutive viral passages on MDCK cells. 4'-FlU was synthesized from 5'-(3-chlorobenzoyloxy)-2',3'-di-O-acetyl-4'-fluorouridine and 4N ammonia in methanol as previously described [24], authenticated through elemental analysis, and stored as dry powder. Working aliquots were dissolved in DMSO or 10 mM sodium citrate with 0.5% Tween 80 for *in vitro* and *in vivo* studies, respectively.

## Virus titration

Virus samples were serially diluted (10-fold starting at 1:10 initial dilution) in serum-free DMEM supplemented with 0.5% Trypsin (Gibco). Serial dilutions were added to MDCK cells seeded in 96-well plate at $1\times10^4$ cells per well 24 hours before infection. Infected plates were incubated for 72 hours at 37˚C with 5% $CO_2$, followed by transfer of culture supernatants to suspension of chicken red blood cells and scoring of wells based on hemagglutination activity. A/Viet Nam/1203/2004 (H5N1) virus and samples were titered by plaque assay; serially diluted 10-fold in DMEM with 2% FBS, added to MDCK cells in 12-well plates and overlaid with DMEM+2% FBS with 1.2% Avicel microcrystalline cellulose. Plates were incubated for 48 hours at 37˚C with 5% $CO_2$, then washed, fixed with methanol:acetone (80:20), and counterstained with crystal violet to visualize plaques.

## Dose-response assays

Active concentrations ($EC_{50}$ and $EC_{90}$) of 4'-FlU were calculated through 4-parameter variable slope regression modeling of dose-response assay data, derived from progeny virus yield ($TCID_{50}$) assays, luciferase reporter virus assays, or minigenome assays as specified. To test compound potency, 4-FlU was in all dose-response assays assessed in 3-fold serial dilutions, each tested in 3 independent repeats (biological repeats), with starting concentrations of 20 μM 4'-FlU. Control wells received vehicle (DMSO) volume equivalents. Dose-response assays on MDCK cells and 2D HAE cultures were carried out in 96-well plate format ($1.5 \times 10^4$ cells/well) or 24-well plate format ($1.5 \times 10^5$ cells/well), respectively. MDCK cells were infected in serum-free DMEM supplemented with 0.5% TPCK-treated trypsin (Gibco) with an MOI of 0.02 $TCID_{50}$ units/cell, HAE cultures were infected in BronchiaLife media (LifeLine) supplemented with 0.75 μg/ml TPCK-treated trypsin with an MOI of 1.0 $TCID_{50}$ units/cell. Culture supernatants were removed 24 (HAE cultures) or 36–48 hours (MDCK cells) after infection and viral titers determined through $TCID_{50}$ titration. For luciferase reporter viruses, relative luciferase activity was read *in situ* using a BioTek Synergy H1 multi-mode reader and signals normalized according to the formula % relative activity = ($RLU_i$−$RLU_{min}$)/($RLU_{max}$−$RLU_{min}$) × 100, with $RLU_i$ specifying sample luciferase value, $RLU_{min}$ specifying values from equally transfected, vehicle-treated cells that did not receive the PB2

subunit encoding plasmid, and $RLU_{max}$ specifying values from fully transfected, vehicle-treated cells.

## Primary human airway epithelium cultures grown at air-liquid interface

Primary HAEs ($3 \times 10^5$/well) at passage <3 were seeded on 6.5 mm 0.4 µm pore size polyester transwell inserts (Corning Costar Transwell) and differentiated for at least 21 days at air-liquid interface using Air-Liquid Interface Differentiation Medium (LifeLine Cell Technology). Developing HAEs were washed apically at least once per week to remove mucus. Transepithelial/transendothelial resistance (TEER) was measured every 24 hours for 3 days after infection using an EVOM volt/ohm meter coupled with STX2 electrode (World Precision Instruments). For infection, virus (5,000 $TCID_{50}$ units/transwell) was added apically. Compound was administered to the basolateral chamber, starting with a highest concentration of 0.5 µM 4'-FlU in dose-response assays, and exchanged once daily. Each concentration was tested in three independent replicates (biological repeats). Apically shed virus titers were determined through $TCID_{50}$ titration on MDCK cells as described above.

## Minigenome assay

293T cells were seeded and transfected with 1.0 µg of luciferase-encoding reporter plasmid [26] and 0.5 µg each of PB1, PB2, NP and PA expression plasmids derived from A/California/07/2009 (H1N1), A/Viet Nam/1203/2004 (H5N1), and A/Anhui/1/2013 (H7N9), respectively. Three hours post transfection, 4'-FlU or vehicle (DMSO) volume equivalents were added in 3-fold serial dilution series starting at 20 µM 4'-FlU concentration, followed by incubation for 30 hours at 37°C and determination of luciferase activity using a BioTek Synergy H1 multimode plate reader. Relative activities were calculated according to the formula % relative activity = $(RLU_i - RLU_{min})/(RLU_{max} - RLU_{min}) \times 100$, with $RLU_i$ specifying sample luciferase value, $RLU_{min}$ specifying values from equally transfected, vehicle-treated cells that did not receive the PB2 subunit encoding plasmid, and $RLU_{max}$ specifying values from fully transfected, vehicle-treated cells.

## Recombinant IAV RdRP expression and purification

A/CA/07/2009 (H1N1)-derived PA, PB1, and PB2 expression plasmids were subcloned into the pFastbac Dual plasmid expression vector (Invitrogen) with polyhedrin promoter control. An TEV protease-cleavable N-terminal hexahistidine-tag was added to the PA subunit to facilitate purification of the complex through affinity chromatography. The individual plasmids were transformed in DH10bac cells and the resulting bacmid transfected into SF9 cells. After virus recovery, SF9 cells were infected with the three individual viruses in the ratio of 1:1:2. Cells were harvested after 72 hours and lysed in lysis buffer (Tris 50 mM pH 7.4, NaCl 300 mM, NP 40 0.5%, protease inhibitor cocktail, glycerol 10%, TCEP 0.5 mM) on ice for two hours, followed by sonication. Clarified supernatant was incubated at 4°C for 1 hour with Ni-NTA beads which were pre-equilibrated with Tris 50 mM pH 7.4, NaCl 300 mM, NP 40 0.2%, glycerol 10%, TCEP 0.5 mM containing 20 mM imidazole. After extensive washing with Tris 50 mM pH 7.4, NaCl 300 mM, NP 40 0.2%, glycerol 10%, TCEP 0.5 mM containing 20–50 mM imidazole, protein complexes were eluted in buffer Tris 50 mM pH 7.4, NaCl 300 mM, NP 40 0.2%, glycerol 10%, TCEP 0.5 mM containing 250 mM imidazole. Eluted protein fractions were pooled and loaded on heparin HP columns, which were flushed with buffer A (Tris 50 mM pH 7.4, NaCl 250 mM, glycerol 10%), then 0–100% of buffer B (Tris 50 mM pH 7.4, NaCl 1M, glycerol 10%) diluted in buffer A. The polymerase complex peak was observed in ~80% buffer B, which corresponds to approximately 900 mM NaCl in the buffer. Peak

fractions were collected separately and dialyzed against buffer C (Tris/Cl 50 mM pH 7.4, NaCl 150 mM, glycerol 10%) in dialyzing cassettes at 4˚C overnight. Purified protein complexes were harvested, aliquoted and stored at -80˚C until use.

### *In vitro* RdRP primer extension assay

*In vitro* primer extension assays were carried out essentially as described [24,61]. Briefly, RdRP assay reactions were set up in 20 mM Tris-HCl pH 7.4, 10% glycerol, 1 mM TCEP, 3 mM MnCl2, 2 µM RNA template, 100 µM RNA primer, and nucleotides and 1 µCi of $\alpha^{32}$P-labelled ATP (Perkin-Elmer) as specified in the figure legends, followed by incubation at 30˚C for 10 minutes. Polymerase complexes were added to the reaction and incubation continued for 1 hour at 30˚C. Reactions were stopped with 1 volume of deionized formamide containing 25 mM ethylenediaminetetraacetic acid (EDTA), followed by a denaturation step at 95˚C. Amplicons were fractionated on 20% polyacrylamide gels with 7M urea, using a Tris-Borate-EDTA electrophoresis buffer. Amplicons were visualized by autoradiography, using a storage phosphor screen BAS IP MS 2040 E (GE Healthcare Life Sciences) and Typhoon FLA 7000 (GE Healthcare Life Sciences) imager. Densitometry analysis was performed using FIJI 2.1.0. To determine enzyme kinetics ($V_{max}$ and $K_m$), Michaelis-Menten equation was applied in Prism 9.4.1 (GraphPad).

### Pharmacokinetic property and tissue distribution profiling in mice

Male and female C57BL/6J mice were rested for 3 days prior to study start. Then, animals were gavaged with 1.5 mg/kg bodyweight 4'-FlU in 10 mM sodium citrate with 0.5% Tween 80, final gavage volume 200 µl. Blood samples (150 µl) were taken at 90 minutes, 3 hours and 12 hours after dosing, cleared (2,000 rpm, 5 minutes, 4˚C), and extracted plasma stored at -80˚C. Organ samples (lung, brain, spleen, heart, kidney and liver) were harvested 12 hours after dosing and snap-frozen in liquid nitrogen. Animal tissues were homogenized with 70% acetonitrile in water that included internal standards. Animal plasma and tissue concentrations of 4'FlU and 4'-FlU-TP were measured by a qualified LC/MS/MS method in MRM mode on a QTRAP 5500 (Sciex, Santa Clara, CA, USA) instrument.

### Intranasal infection of mice

Female or male mice (5–8 weeks of age, Balb/c, RAG1 KO or IFNar KO) were purchased from Jackson Laboratories. Upon arrival, mice were rested for at least 3 days, then randomly assigned to study groups and housed under ABSL-2 or ABSL-3 conditions for infections with A/CA/07/2009 (H1N1) recombinants or A/VN12/2003 (H5N1), respectively. Bodyweight was determined twice daily, body temperature determined once daily rectally. For infections, animals were anesthetized with isoflurane or isoflurane/ketamine (H5N1 challenge only), followed by intranasal inoculation with $2.5 \times 10^1$ TCID$_{50}$ units/animal of A/CA/07/2009 (H1N1), $1 \times 10^3 – 1 \times 10^5$ TCID$_{50}$ units/animal of GFP or nanoLuc expressing reporter virus versions of A/CA/07/2009 (H1N1), or $6 \times 10^1$ of A/VN12/2003 (H5N1). In all cases, virus inoculum was diluted in 50 µl and administered in amounts of 25 µl per nare. Animals were euthanized and organs harvested at predefined time points or when animals reached humane study endpoints.

### 4'-FlU anti-IAV efficacy studies in mice

Mice were inoculated intranasally with IAV stocks as described, followed by treatment with 4'-FlU at specified dose levels and starting time points through oral gavage. Unless specified otherwise, treatment was continued q.d. Each individual study contained animals receiving equal

volumes of vehicle through oral gavage. At study end point, lung tissue was harvested for virus titration or histopathology assessment. For tissue fixation, lungs were perfused using 10% neutral-buffered formalin [10% formalin, $NaH_2PO_4$ (4 g/liter), and $Na_2HPO$ (6.5 g/liter)], dissected, and fixed for 24 hours. Formalin-fixed lungs were transferred to 70% EtOH after 72 hours, embedded in paraffin, sectioned (5-μm thickness), and stained with hematoxylin and eosin. Slides were evaluated by a licensed pathologist. A pathology score range of 0 to 3 was used to evaluate tissue damage. For analysis of proinflammatory cytokine levels through Bioplex, lungs of euthanized mice were lavaged with 1 ml sterile PBS to extract BALFs, and cleared BALF analyzed against the Standard Th17-panel (Bio-Rad) according to the manufacturer's instructions.

### Efficacy studies in ferrets

Female ferrets (6–10 months of age) were purchased from Triple F Farms. Upon arrival, ferrets were rested for 1 week, then randomly assigned to study groups and housed in groups of three animals under ABSL-2 conditions. In some studies, animals received a temperature sensor for continued telemetric measurement of body temperature. Otherwise, body temperature was determined rectally once daily. Dexmedetomidine/ketamine anesthetized animals were inoculated intranasally with $1\times10^5$ $TCID_{50}$ units of A/CA/07/2009 (H1N1) in a volume of 500 μl per nare. Nasal lavages were performed twice daily using 1 ml of PBS containing 2× antibiotics-antimycotics (Gibco). Treatment with 4'-FlU through oral gavage followed a q.d. regimen. Ferrets were terminated four days post infection and BALF and organ (lungs, turbinates and tracheas) samples extracted for virus titrations.

### Ferret transmission studies

Female ferrets (3–8 months of age) were anesthetized with dexmedetomidine/ketamine and infected intranasally with $1\times10^5$ $TCID_{50}$ units of A/CA/07/2009 (H1N1) as before. Nasal lavages were performed every 12 hours, bodyweight and temperature determined once daily. Treatment of source animals was initiated 12 or 24 hours after infection at a dose of 2mg/kg bodyweight, and continued q.d. for three doses total. Starting day 2.5 after infection, uninfected and untreated contact animals were added to the infected and treated source ferrets to allow direct-contact transmission, and co-housing continued until termination of the source ferrets on study day 4. Sentinels were subjected to once daily nasal lavages and monitored until study day 10 or, when infectious particles were detected in lavages of contacts of 4'-FlU-treated source animals, study day 14.

### Virus titration from tissue samples

Organs were weighed and homogenized in 300 μl PBS using a beat blaster, set to 3 cycles of 30 seconds each at 4°C, separated by 1-minute rest periods. Homogenates were cleared (10 minutes at $20,000 \times g$ and 4°C), and cleared supernatants stored at -80°C until virus titration. Viral titers were expressed as $TCID_{50}$ units per gram input tissue. For A/VN12/2003 (H5N1), lungs were homogenized using a TissueLyzer (Qiagen), clarified by centrifugation, supernatants aliquoted, and then stored at -80°C. A/VN12/2003 (H5N1) virus titer was determined by plaque assay and expressed as PFU per ml of lung homogenate.

### Infection of mice with aerosolized A/CA/07/2009 (H1N1)

An Aeroneb (Kent Scientific) system was used to infect mice (female Balb/c, 6–8 weeks of age) with aerosolized IAV. Conscious animals were transferred to an aerosol chamber placed on a

heat pad adjusted to 37˚C to prevent condensation, connected to the nebulizer unit (Aeroneb with palladium mesh) and air pump with airflow restrictor. For condition-finding experiments, the nebulizer chamber was loaded with virus inoculum diluted to $1\times10^3$–$1\times10^5$ $TCID_{50}$ A/CA/07/2009 (H1N1) per ml in sterile PBS, airflow adjusted to 1 l/minute, and animals continuously exposed for 3, 10, or 30 minutes. For treatment studies, conditions were adjusted to $1\times10^5$ $TCID_{50}$/ml in sterile PBS, 10 minutes exposure time, and 1 l/minute airflow.

## Re-infection of 4'-FlU-experienced mice with A/CA/07/2009 (H1N1)

Mice infected with aerosolized A/CA/07/2009 (H1N1) as above were treated q.d. with 4'-FlU or vehicle as specified, and monitored for 10 weeks or until humane endpoints were reached. Blood was collected 4 and 8 weeks after infection and A/CA/07/2009 (H1N1)-neutralizing antibody titers determined. Treated recoverees were reinfected with homotypic, aerosolized A/CA/07/2009 (H1N1) 10 weeks after the original infection along with a fresh set of naïve animals, followed by monitoring for an additional 14 days and final blood collection at study end. No treatment was administered after reinfection.

## Neutralizing antibody titers in mice

Plasma was prepared from blood samples (2,000 rpm, 10 minutes, 4˚C) and stored at -80˚C until analysis. Heat-inactivated plasma was incubated with 100 $TCID_{50}$ units of A/CA/07/2009 (H1N1) for 90 minutes, followed by serial dilution and transfer to MDCK cells, and plates incubated for 3 days at 37˚C. Remaining infectivity was visualized through hemagglutination activity on chicken red blood cells.

## Statistical analysis

For statistical analysis of studies consisting of only two groups, unpaired two-tailed t-tests were applied. When comparing more than two study groups, 1-way analysis of variance (ANOVA) or 2-way ANOVA with multiple comparison *post hoc* tests as specified were used to assess statistical difference between samples. Statistical analyses were carried out in Prism version 9.4.1 (GraphPad). The number of individual biological replicates (n values) and exact P values are shown in the figures when possible. The threshold of statistical significance (α) was set to 0.05. All quantitative source data and statistical analyses are shown in S1 and S2 data files, respectively.

## Supporting information

**S1 Fig. 4'-FlU activity in undifferentiated and differentiated HAE cultures. a)** 4'-FlU antiviral activity in 2D HAE cultures. Dose-response assays of 4'-FlU against a A/WSN/33 (H1N1) nano-luciferase reporter virus (WSN-nanoLuc) on undifferentiated HAEs derived from 10 different healthy donors, five male and five female. Lines represent 4-parameter variable slope regression models; symbols show data means ± SD; n = 6. $EC_{50}$ and $EC_{90}$ concentrations are specified. **b)** Dose-response assay of 4'-FlU against pdmCa09 on a well-differentiated HAE culture grown at air-liquid interface. Cultures were infected apically, compound was added to the basolateral chamber; line intersects, and symbols show, geometric means ± SD of apically shed virus (n = 3). $EC_{50}$ and $EC_{90}$ values based on 4-parameter variable slope regression modeling are shown. **c)** TEER measurements of HAE cultures from (b). Symbols show individual measurements (individual transwells); lines intersect, and symbols show, data means ± SD. (TIF)

**S2 Fig. Urea-PAGE fractionation (uncropped) of RNA transcripts. a)** Template and primer used in the reaction. **b-e)** Independent repeats of primer extension assay by IAV polymerase. Dashed rectangle shows insert presented in Fig 1E. Replicates were included in quantitations presented in Fig 1E and 1F.
(TIF)

**S3 Fig. Clinical signs in 4'-FlU-treated mice used to establish treatment paradigms. a-h)** Body weight measurements taken twice daily (a, c, e, g) and rectal body temperature determined once daily (b, d, f, h). Dashed horizontal line specifies humane endpoint of 20% body weight loss; lines intersect, and symbols show, data means ± SD; n values are specified in main Fig 2. Results are shown for lowest efficacious dose (a-b), latest onset of efficacious treatment (c-d), virus clearance (e-f) and minimal number of doses required (g-h) studies.
(TIF)

**S4 Fig. Clinical signs in 4'-FlU treated ferrets. a)** Continuous telemetric assessment of body temperature of ferrets in the 4'-FlU efficacy study. **b-e)** Once-daily body weight (b, d) and rectal body temperature (c, e) of ferrets involved in the pdmCa09 transmission studies. Lines intersect, and symbols show, data means ± SD. Dashed lines in (a,c,e) represent onset of fever; dashed lines in (b,d) specify predefined endpoint.
(TIF)

**S5 Fig. Clinical signs in 4'-FlU-treated mice used in immune response-mitigation study. a-b)** Body weight measurements taken twice daily (a), and rectal body temperature determined once daily (b). Lines intersect, and symbols show, data means ± SD; n values are specified in Fig 4. Dashed line in (a) specifies predefined endpoint.
(TIF)

**S6 Fig. Changes in select BALF cytokine levels in infected and 4'-FlU-treated mice.** IFNy, IL-1b, IL-10, and IL-17a levels present in BALF of animals treated as in Fig 4A are shown, calculated relative to levels at time of infection. Lines intersect, and symbols show, data medians with 95% CI; 2-way ANOVA with Tukey's *post hoc* test; P values are specified.
(TIF)

**S7 Fig. Clinical signs of mice involved in mitigation of histopathology study. a-b)** Body weight measurements taken twice daily (a), and rectal body temperature determined once daily (b). Lines intersect, and symbols show, data means ± SD. Dashed line in (a) specifies predefined endpoint.
(TIF)

**S8 Fig. Photomicrographs of lung tissues.** Lungs were extracted on study days 5 or 21 (infected and treated animals only) of all animals examined in this study; tissue section were H&E stained; dashed rectangle shows sections presented in Fig 4E.; magnification 10×; scale bar 100 μm; Br, bronchiole; Bl or arrowhead, blood vessel.
(TIF)

**S9 Fig. Detailed histopathological scores of all animals examined in this study.** Individual clinical scores for alveolitis, bronchiolitis, perivascular cuffing (PVC), vasculitis, interstitial pneumonia (IP), and pleuritis. Columns show data medians, symbols represent individual animals; 1-way ANOVA with Tukey's *post hoc* test; p values are specified; n values are specified in Fig 4F.
(TIF)

**S10 Fig. Clinical signs in pdmCa09-infected immunocompromised mice treated with 4'-FlU. a-c)** Shown are body weight measurements taken twice daily (a, c), and rectal body temperature (b, d) determined once daily. Lines intersect, and symbols show, data means ± SD; n values are specified in Fig 5. Results are shown for RAG1 KO (a-b) and IFNAR1 KO (c-d) animals. Dashed lines in (a,c) specify predefined endpoint.
(TIF)

**S11 Fig. Set-up of the Aeroneb nebulizer infection system. a-b)** Schematic representation of the system (a) and photograph of the assembly (b).
(TIF)

**S12 Fig. Variation of duration of animal exposure in the aerosolization chamber. a-b)** Body weight measurements taken twice daily (a), and rectal body temperature (b) determined once daily. Lines intersect, and symbols show, data means ± SD. **c)** Lung viral titers determined 4 days after infection. Columns represent geometric means ± SD; symbols show individual animals (n = 4). Dashed line in (a) specifies predefined endpoint.
(TIF)

**S13 Fig. Variation of pdmCa09 inoculum amount for infection in the aerosolization chamber. a)** Body weight measurements taken twice daily. Lines intersect, and symbols show, data means. **b)** Lung viral titers determined 4 days after infection. Columns represent data mean with range; symbols show individual animals (n = 2). Dashed line in (a) specifies predefined endpoint.
(TIF)

**S14 Fig. 4'-FlU efficacy confirmation after infection with aerosolized inoculum virus. a)** Efficacy study schematic; n values are specified. **b-c)** Body weight measurements of animals from (a) taken twice daily (a), and rectal body temperature (b) determined once daily. Lines in (b-c) intersect, and symbols show, data means ± SD. **d)** Lung viral titers determined 4 days after infection. Columns represent geometric means ± SD, symbols show individual animals; 1-way ANOVA with Dunnett's *post hoc* test; P values are specified. Dashed line in (d) specifies limit of detection.
(TIF)

**S15 Fig. Clinical signs of mice infected and reinfected with pdmCa09. a-d)** Body weight measurements taken twice daily (a, c), and rectal body temperature determined once daily (b, d). Lines intersect, and symbols show, data means ± SD. Results are shown for animals after the original infection (a-b) and after homotypic reinfection (c-d). Dashed lines in (a,c) specify predefined endpoint.
(TIF)

**S1 Data. All quantitative source data generated in this study.**
(XLSX)

**S2 Data. All statistical analyses performed for this study.**
(XLSX)

## Acknowledgments

We thank AC Lowen and SM Kang for IAV and IBV virus isolates and the Georgia State University Department of Animal Resources for Assistance.

## Author Contributions

**Conceptualization:** Richard K. Plemper.

**Data curation:** Carolin M. Lieber, Megha Aggarwal, Robert M. Cox.

**Formal analysis:** Carolin M. Lieber, Megha Aggarwal, Julien Sourimant, Alexander A. Kolykhalov, Stephen M. Tompkins, Oliver Planz, Kaori Sakamoto, Richard K. Plemper.

**Funding acquisition:** Richard K. Plemper.

**Investigation:** Carolin M. Lieber, Megha Aggarwal, Jeong-Joong Yoon, Robert M. Cox, Hae-Ji Kang, Julien Sourimant, Mart Toots, Scott K. Johnson, Cheryl A. Jones, Zachary M. Sticher, Manohar T. Saindane, Kaori Sakamoto, Richard K. Plemper.

**Methodology:** Manohar T. Saindane.

**Project administration:** Michael G. Natchus, Richard K. Plemper.

**Supervision:** Alexander A. Kolykhalov, Stephen M. Tompkins, George R. Painter, Michael G. Natchus, Richard K. Plemper.

**Validation:** Richard K. Plemper.

**Writing – original draft:** Carolin M. Lieber, Richard K. Plemper.

**Writing – review & editing:** Robert M. Cox, Julien Sourimant, Stephen M. Tompkins, George R. Painter, Michael G. Natchus, Kaori Sakamoto.

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
