## [Decision Letter · Decision Letter 0]

28 Jan 2023

Dear Dr. Plemper,

Thank you very much for submitting your manuscript "4’-Fluorouridine mitigates lethal infection with pandemic human and highly pathogenic avian influenza viruses" for consideration at PLOS Pathogens. As with all papers reviewed by the journal, your manuscript was reviewed by members of the editorial board and by several independent reviewers. The reviewers appreciated the attention to an important topic.  They have requested that a number of issues be addressed, including adding statistical analyses, more carefully considering what HAE cultures represent, and discussing several observations in more detail. I ask that you address these concerns as you revise your manuscript. Based on the reviews, we are likely to accept this manuscript for publication, providing that you modify the manuscript according to the review recommendations.

Sincerely,

Rebecca Ellis Dutch

Pearls Editor

PLOS Pathogens

Matthias Schnell

Section Editor

PLOS Pathogens

Kasturi Haldar

Editor-in-Chief

PLOS Pathogens

orcid.org/0000-0001-5065-158X

Michael Malim

Editor-in-Chief

PLOS Pathogens

orcid.org/0000-0002-7699-2064

Thank you for your submission. The reviewers appreciated the significance of the work and the care in which the experiments and the manscript writing were approached. They have requested that a number of issues be addressed, including adding statistical analyses, more carefully considering what HAE cultures represent, and discussing several observations in more detail. I ask that you address these concerns as you revise your manuscript.

Reviewer Comments (if any, and for reference):

Reviewer's Responses to Questions

**Part I - Summary**

Reviewer #1: The work presented in this manuscript evaluated the efficacy of the 4’-Fluorouridine derivative compound, which was previously shown to be highly effective against SARS-CoV-2 and RSV, on influenza A and B viruses in vitro and in vivo (with two animal models. The results demonstrate that the compound is highly potent against highly pathogenic influenza viruses including avian flu H5N1. It was shown that the compound is still effective for the mouse survival 60 hours post viral infection, and it can block direct viral transmission using a ferret model. The experiments were carefully designed and performed, and the manuscript well written. This compound has potential to be developed as a new anti-flu drug.

Reviewer #2: The manuscript describes in vitro and in vivo efficacy of 4’-fluorouridine (4FIU) against a number of influenza A virus strains. This drug had previously been shown to have activity against RSV and SARS-CoV-2 by functioning to prematurely terminate RNA synthesis, and this mechanism of action was shown to be even more potent against influenza. The study’s strengths are that it combines a wide range of in vitro assays with a comprehensive analysis of drug efficacy in two animal species, in addition to immunocompromised mice.

Reviewer #3: In this manuscript, the authors tested a small molecule (4’-fluorouridine , 4’-FIU) as an antiviral for the influenza virus (IV). The work encompasses both in vitro, ex vivo, in vivo work and includes an attempt to assess the mechanism of action.

Some in vitro experiments are rigorously presented, with exceptions noted below.

The ex vivo data in human airway epithelium (HAE) are a good addition but the data presented are incomplete and not adequately described (see below). Additionally, the word “organoids” is misleading and is generally used for different kinds of engineered tissues. This should be corrected (not just the semantics but the actual consideration of what these tissues reflect) or alternatively, the experiment should be conducted on lung tissues.

The in vivo data are stronger and overall suggest that the 4’-FIU is effective even when administered 24 hours after initial infection. This is in line with the efficacy of the neuraminidase inhibitors currently approved and in use for IV. The authors claim that the advantage of their strategy is the high barrier to resistance, but no data are presented in this regard. The authors should either present data on the viral evolution of their small molecule compared to the other available antivirals to substantiate their claim or remove this claim.

The 4’-FIU small molecule has been shown to have broad-spectrum activity vs. other viruses like SARS-CoV-2 and RSV. For this reason, the novelty of the work is modest.

Specific comments:

-Fig 1b presents data from in vitro infection with several strains however no details of the experiment are presented.

-Fig 1b shows a wide range of efficacy with EC90 varying from 0.06 to 5.347uM. The authors should comment on these differences. Is the difference due to the inherent partial resistance of the different strains? Since the authors have an assay to assess the activity (Fig.1d to f), the molecular underpinnings for the difference in activity should be tested. Otherwise, concerns are raised about the lack of understanding of the basis for these differences.

-Fig 1c is missing crucial details about the experiment (time points, viral dosage, etc.). Since the mouse-adapted influenza virus is self-extinguishing in this tissue and human strains totally destroy the tissue, information about the TEER and images of the infected HAE during the course of infection should be provided to evaluate the protective effect of the 4’-FIU over time.

-Fig 1b and Fig 1c : there is a major difference in activity vs. the same virus (EC90 0.7uM in MDCK and 0.074uM in HAE). In HAE it seems that the potency is much higher than in MDCK. The authors should comment on this difference which points to an underlying mechanism that needs to be explained.

**Part II – Major Issues: Key Experiments Required for Acceptance**

Reviewer #1: None

Reviewer #2: Figure 1B – is it surprising that the EC50s of the drug for IBV and the duck and swine derived IAVs is 10-50 fold higher than that seen with the human IAVs tested?

Reviewer #3: The authors claim that the advantage of their strategy is the high barrier to resistance, but no data are presented in this regard. The authors should either present data on the viral evolution of their small molecule compared to the other available antivirals to substantiate their claim or remove this claim.

Fig 1b shows a wide range of efficacy with EC90 varying from 0.06 to 5.347uM. The authors should comment on these differences. Is the difference due to the inherent partial resistance of the different strains? Since the authors have an assay to assess the activity (Fig.1d to f), the molecular underpinnings for the difference in activity should be tested. Otherwise, concerns are raised about the lack of understanding of the basis for these differences.

Fig 1b and Fig 1c : there is a major difference in activity vs. the same virus (EC90 0.7uM in MDCK and 0.074uM in HAE). In HAE it seems that the potency is much higher than in MDCK. The authors should comment on this difference which points to an underlying mechanism that needs to be explained.

Fig 1c is missing crucial details about the experiment (time points, viral dosage, etc.). Since the mouse-adapted influenza virus is self-extinguishing in this tissue and human strains totally destroy the tissue, information about the TEER and images of the infected HAE during the course of infection should be provided to evaluate the protective effect of the 4’-FIU over time.

**Part III – Minor Issues: Editorial and Data Presentation Modifications**

Reviewer #1: 1. Add statistics/significance to the survival curves in the figures

2. Is there an explanation for the discrepancy in the potency between the flu isolates in Figure 1B?

Reviewer #2: Figure 1C – given the disparate activity of 4FIU against IBV and swine/duck IAV, it would be important to ensure that the drug was still active in the same dosing range on the HAE organoid cultures.

HPAI experiments – its not clear why the authors boosted the drug dose to 5 mg/kg in the HPAI experiments, when 2 mg/kg was used in experiments with 2009 H1N1.

Reviewer #3: Fig 1b presents data from in vitro infection with several strains however no details of the experiment are presented.

PLOS authors have the option to publish the peer review history of their article (what does this mean?). If published, this will include your full peer review and any attached files.

Reviewer #1: **Yes: **Lijun Rong

Reviewer #2: No

Reviewer #3: No

Figure Files:

Data Requirements:

Reproducibility:

References:

---

## [Editor Report · Decision Letter 1]

28 Mar 2023

Dear Dr. Plemper,

Thank you very much for submitting your manuscript "4’-Fluorouridine mitigates lethal infection with pandemic human and highly pathogenic avian influenza viruses" for consideration at PLOS Pathogens. As with all papers reviewed by the journal, your manuscript was reviewed by members of the editorial board and by several independent reviewers. The reviewers appreciated the attention to an important topic. Based on the reviews, we are likely to accept this manuscript for publication, providing that you modify the manuscript according to the review recommendations.

Sincerely,

Matthias Johannes Schnell, PhD

Section Editor

PLOS Pathogens

Matthias Schnell

Section Editor

PLOS Pathogens

Kasturi Haldar

Editor-in-Chief

PLOS Pathogens

orcid.org/0000-0001-5065-158X

Michael Malim

Editor-in-Chief

PLOS Pathogens

orcid.org/0000-0002-7699-2064

Reviewer Comments (if any, and for reference):

Figure Files:

Data Requirements:

Reproducibility:

References:

---

## [Editor Report · Decision Letter 2]

3 Apr 2023

Dear Dr. Plemper,

We are pleased to inform you that your manuscript '4’-Fluorouridine mitigates lethal infection with pandemic human and highly pathogenic avian influenza viruses' has been provisionally accepted for publication in PLOS Pathogens.

Best regards,

Matthias Johannes Schnell, PhD

Section Editor

PLOS Pathogens

Matthias Schnell

Section Editor

PLOS Pathogens

Kasturi Haldar

Editor-in-Chief

PLOS Pathogens

orcid.org/0000-0001-5065-158X

Michael Malim

Editor-in-Chief

PLOS Pathogens

orcid.org/0000-0002-7699-2064
---

## [Editor Report · Acceptance letter]

12 Apr 2023

Dear Dr. Plemper,

We are delighted to inform you that your manuscript, "4’-Fluorouridine mitigates lethal infection with pandemic human and highly pathogenic avian influenza viruses," has been formally accepted for publication in PLOS Pathogens.

Best regards,

Kasturi Haldar

Editor-in-Chief

PLOS Pathogens

orcid.org/0000-0001-5065-158X

Michael Malim

Editor-in-Chief

PLOS Pathogens

orcid.org/0000-0002-7699-2064